# Perceived Collective School Efficacy Mediates the Organizational Justice Effect in Teachers’ Subjective Well-Being

**DOI:** 10.3390/ijerph191710963

**Published:** 2022-09-02

**Authors:** Camilo Herrera, Javier Torres-Vallejos, Jonathan Martínez-Líbano, Andrés Rubio, Cristian Céspedes, Juan Carlos Oyanedel, Eduardo Acuña, Danae Pedraza

**Affiliations:** 1Doctoral Candidate in Health, Well-Being and Quality of Life, Faculty of Education and Psychology, University of Girona, 17003 Girona, Spain; 2Faculty of Education and Social Sciences, Andres Bello University, Santiago 7591538, Chile; 3Faculty of Economics and Business, Andres Bello University, Santiago 7591538, Chile; 4Faculty of Psychology, Diego Portales University, Santiago 8370076, Chile; 5Faculty of Social Sciences, University of Chile, Santiago 7800003, Chile

**Keywords:** collective efficacy, organizational justice, teachers, Chile, subjective well-being

## Abstract

Trust and team communication are central aspects for the achievement of both individual and common goals, which affect not only work efficiency but also the well-being of its members. In addition, organizational justice could affect these indicators, as well as the perception of collective efficacy within organizations, in this case, schools. The purpose of this study was to analyze the effect of organizational justice on teachers’ subjective well-being, and how this is affected/mediated by collective efficacy. We worked with a sample of 693 teachers across Chile. Multiple mediation analysis was carried out, where the latent variables of the study were estimated (subjective well-being, organizational justice, and two dimensions of collective efficacy). The results indicate that there is full mediation of the collective efficacy dimensions between the predictor–criterion relationship. Our findings allow us to hypothesize that perceptions of collective efficacy are central to explaining well-being as an intrinsic factor.

## 1. Introduction

Teachers are an essential part of school life as they are not only responsible for students’ learning but also play a relevant role in the process of children’s emotional literacy and their integral formation [1,2]. Teachers are also recognized as agents of change, playing an important role in the emotional adaptation of students to new scenarios [3]. To this end, it is important to be concerned not only with making the school community work but also with the mental health and well-being of all members of the community [4]. The implementation of Law 20.903, which establishes the Teacher Professional Development System and is one of the cornerstones of the present Educational Reform, has recently resulted in certain improvements to the educational system in Chile, including higher compensation and a culture of accountability. This system establishes changes to offer remedies and act in situations involving teacher professionalism, needs for performance support, and evaluation [5]. Although it does not significantly improve the working environment or workplace organization, this new professional development system aims to strengthen basic teacher training, among other things [6]. Since the late 1980s and the mid-1990s, the OECD has promoted these educational changes [7], and they have also occurred in Latin America, particularly from 2002 onwards [8], especially influencing countries such as Mexico and Chile [8], with a consequently significant impact on teachers’ careers and training [9]. This law’s intriguing proposal to cut teaching hours is said to benefit teachers’ rest time, time for interaction and teamwork with peers, levels of educational management, reflection on their practices, relationships with students, innovation, and professional growth, many of which are positively correlated with teachers’ subjective well-being [10,11,12,13].

In this line, subjective well-being is defined as a general assessment that people make about their lives, the events that happen to them, and the circumstances in which they live [14,15]. Research has shown that subjective well-being is positively associated with individual positive mental health and plays a key role in reducing the devastating effects of mental illnesses, anonymity, social marginality, and poverty [16,17,18]. In this regard, research on Chilean teachers’ quality of life and well-being is limited, but findings show that they are less satisfied than teachers in industrialized nations, and public policy should address this [19,20].

The idea of social well-being extends the notion of subjective well-being by placing the individual in context and in relation to other individuals in the social and educational milieu. [20,21]. Research has shown that a good quality of community social support can explain the sense of well-being in the school community [22]. Schools that have a relational system reinforcing the formation of bonds can have a direct impact on reducing the level of stress in individual mental health problems, and at a social level, in violent social climates [23]. Positive relationships among the members of the school community also affect learning processes [24,25,26,27].

Thus, Yañez states that “people always need to trust, both in affective and work and business relationships” [28] (p. 44), meaning that trust has been considered a fundamental element in the social development of each person and the construction of citizenship. Building trust in school requires the creation of certain conditions, and at the same time, this process generates an impact on the institution in different aspects of the work environment [29]: honesty, benevolence, sincerity and/or openness, trustworthiness, and competence [29]. Regarding trust building, Hoy and Tarter state that the development of trust in an institution can promote teamwork, reduction of vulnerability, problem-solving, and encourages individuals to risk interdependence and leadership development [30]. All of the above enhance a principal’s ability to shape the mission and influence the behaviors of the institution’s members.

In this line, collective school efficacy is produced when a group of teachers in a school community believe that a joint effort and their set of skills can yield better academic results in their students [31]. Under the vision of Bandura, teachers should be able to use their “conjoint capability to organize and execute the courses of action required to produce given levels of attainment” [32] (p. 477). There is robust evidence of the correlation between well-being and collective efficacy in teachers [33,34].

The working environment also plays an important role in teachers’ well-being, especially when it comes to good relations and the feeling of being appreciated and fairly treated [19,35]. In this line, organizational justice refers to individuals’ perceptions of fairness in organizations [36] and at schools, a fact of great importance, as it is proven to have an impact on teachers’ commitment, well-being, motivation, and behavior, among other constructs [37,38,39,40]. Regarding organizational justice in Chilean school settings, recently, attention has been paid to improving work environments, maintaining good relationships in work teams, and building a climate of trust [41].

The aim of this study is to test the effect of organizational justice on the subjective well-being of teachers and see how the perception of collective efficacy mediates this relationship. The research question of the study is: how does perceived collective school efficacy mediate the effect of organizational justice on the subjective well-being of teachers?

## 2. Materials and Methods

### 2.1. Participants and Procedure

This study used a probabilistic and stratified sample of schools in the different urban zones of the 16 regions of Chile, considering the 2017 National School Enrollment Registry from the Chilean Ministry of Education. The scale was applied through self-report questionnaires during the 2017 school year, within regular school hours, as part of an instrument that included other scales in a larger study. The list of schools in the country published by the Ministry of Education was considered, which was randomized. The schools were contacted by phone with the purpose of inviting them to be part of the study. When the schools agreed to participate, they proceeded to deliver information about the study and schedule visits. If they did not agree to participate, the next school on the list was contacted.

The sample was composed of 693 teachers, and the mean age was 39.4 (SD = 11.8) years. Most of the teachers were from public schools (44.8%) and subsidized schools (42.4%), followed by private ones (11.1%) and another administrative dependency (1.8%). In total, 87.0% of the teachers belonged to primary education (which in the Chilean educational system includes levels ranging from 7 to 14 years), while 13.0% belonged to secondary education (15 to 18 years). According to the Chilean Index of School SES (known as school vulnerability percentage), 50% of the schools catered to students with low SES, 25.4% with medium SES, and 24.6% with high SES.

### 2.2. Instruments

#### 2.2.1. Outcome Variable: Subjective Well-Being

Subjective well-being measured by the Personal Well-being Index for Adults (PWI-A) was originally developed by the International Well-being Group [42] and adapted for the Chilean context [43]. This scale measures subjective well-being by considering seven dimensions of satisfaction (one for each item) and two additional items related to religion or spirituality [44,45], and overall life satisfaction [46]. This 8-item scale is rated on an 11-point scale (0 = “Completely dissatisfied”; 10 = “Completely satisfied”). Confirmatory Factor Analysis showed that the unidimensional model has a good fit to the data (χ²(26) = 114.773 ***; CFI = 0.944; RMSEA = 0.070; SRMR = 0.036). Additionally, it showed good reliability according to McDonald’s ω (ω = 0.901).The Satisfaction With Life Scale (SWLS) [47] demonstrated appropriate psychometric properties in its Spanish version. The items included in this indicator are “in many aspects, my life is close to my ideal”, “my living conditions are excellent”, “I am satisfied with my life”, “so far, I have obtained the things I want in my life”, and “if I could live my life over again, I would change almost nothing”. Responses range from 1 to 7, where 1 means “completely disagree” and 7 means “completely agree”. Confirmatory Factor Analysis showed that the unidimensional model has a proper fit to the data (χ²(4) = 24.879 ***; CFI = 0.978; RMSEA = 0.087; SRMR = 0.019). Additionally, it showed good reliability according to McDonald’s ω (ω = 0.914). The second-order factor for the Subjective well-being variable was tested considering PWI-A and SWLS. Confirmatory Factor Analysis showed a very good fit to the data (χ²(73) = 252.076 ***; CFI = 0.950; RMSEA = 0.059; SRMR = 0.034).

#### 2.2.2. Predictor Variable: Organizational Justice (OJ)

The Organizational Justice Scale (OJS) [30] is measured by a 5-item scale based on behaviors such as faculty trust, faculty trust in the principal, and fair treatment of the school community. It is rated on a 6-point Likert scale from 1 “Strongly disagree” to 6 = “strongly agree”. Confirmatory Factor Analysis showed that the one-factor model has a very good fit to the data (χ²(4) = 17.649 ***; CFI = 0.981; RMSEA = 0.070; SRMR = 0.021). Additionally, it showed good reliability according to McDonald’s ω (ω = 0.878).

#### 2.2.3. Mediator Variables: Collective Efficacy Scale

It was considered an adapted version of the Collective Efficacy Scale Short-Form (CES-SF) developed in 2002 by Goddard [48]. The adapted version of the scale, validated in the Chilean context [49], was used based on its original 21-item version, and the 12 items were rated on a 6-point scale (1 = “strongly disagree”; 6 = “strongly agree”). This proposed scale for Chilean teachers is an 8-item scale comprised of two correlated factors: opportunities for collective efficacy and challenges for collective efficacy. Confirmatory Factor Analysis showed that the two correlated factor model has a very good fit to the data (χ²(18) = 51.547 ***; CFI = 0.951; RMSEA = 0.052; SRMR = 0.035). Reliability analyses showed a McDonald’s ω of 0.769 for the opportunities dimension and 0.609 for the challenges dimension.

### 2.3. Ethical Considerations

Participation in this study was supported by the signature of the researcher and participant in the form of a letter of consent following the regulations of the Ethics Committee of Pontificia Universidad Católica de Valparaíso, Chile, following the Declaration of Helsinki. All participants signed informed consent forms. All questionnaires were administered in the schools where participants worked. This research was approved by the Ethics Committee of Pontificia Universidad Católica de Valparaíso, Chile, under the code BIOEPUCV-H 427-2021.

### 2.4. Analytic Plan

Negatively worded items were reversed before the calculation of later analyses. Descriptive correlation analysis between the study variables was performed. Later, Subjective Well-Being (SWB) was estimated as a latent variable from two measures: PWI-A and SWLS as a dependent variable. Then, predictor and mediator variables were also estimated as latent variables. All direct and indirect effects were estimated by controlling variables of gender, age, and school SES. Mediation models were designed separately to estimate the direct and indirect effects between Organizational Justice items and the SWB scale as a latent variable, provided that both questionnaires included the same items on the topic of this study. Standard errors were calculated using the bootstrap method by 500 re-sampling with maximum likelihood and a 95% confidence interval.

Structural equation modeling (SEM) with maximum likelihood estimation was used to estimate the direct and indirect effects of the models with MPlus version 8.7 [50]. To avoid the loss of cases due to missing values, missing items were completed using the regression imputation method. To impute the value of each missing item of those that made up the scales in question, all the rest of the items that made up each scale were considered as predictor variables, as well as the sociodemographic variables (age, gender, and vulnerability). It was subsequently verified that the imputed values were not outside the defined range for each variable. The scale’s items for the Subjective Well-Being variable had a random distribution, with an average percentage of missing values of 2.44%. The same occurred in the case of the variables Perceived Collective School Efficacy and Organizational Justice, where the average percentage of lost cases for the items that made up each scale corresponded to 7.21% and 8.35%, respectively. The increase in the percentage of lost cases is attributed to the order in which the scales were presented throughout the instrument. Cases that had more than 30% missing items for at least one of the scales that made up the study variables were eliminated. The chi-squared, Bentler Comparative Fit Index (CFI), Steiger–Lind Root Mean Square Error of Approximation (RMSEA), and Standardized Root Mean Square Residual (SRMR) were used as indices to verify the fit of the model. Values above 0.95 in the CFI and below 0.05 in the RMSEA and SRMR were considered excellent adjustments [51,52].

## 3. Results

The results show a significant correlation between the SWB measures and the other interest variables, as well as between them. As expected, both SWB scales are strongly correlated to each other (Table 1). Additionally, reliability coefficients for each variable/scale were also optimal.

Figure 1 show the mediation model used. Prior to this analysis, in order to determine whether our proposed factorial structure was supported by our data, we conducted two confirmatory factor analyses before testing our main hypotheses. We first assumed a single latent variable explaining the variance of all the measures considered in the model (PWI-A, SWLS, OJ, CES with its two dimensions). This model showed a poor fit χ²(296) = 2203.398 ***; CFI = 0.718; RMSEA = 0.096; SRMR = 0.114). Then, we specified a five-factor structure (one for each scale), obtaining an appropriate fit (χ²(286) = 725.102 ***; CFI = 0.935; RMSEA = 0.047; SRMR = 0.040). This analysis provides evidence of the dimensionality of our model that theoretically and empirically distinguishes between the dimensions/scales explored.

For this sample, the model without a mediating variable explained 31.2% of the variance of the SWB latent variable, while the model in Figure 1 explained 23.5%. The mediation model displayed a good fit for this sample (χ²(366) = 1166.301 ***; CFI = 0.921; RMSEA = 0.056; SRMR = 0.048).

The total effect between OJ and SWB was significant (β = 0.330; *p* < 0.001), but this effect disappeared when controlling the mediator variables (β = 0.110; ns.). The standardized indirect effect for “Opportunities” and “Challenges” variables was statistically significant (β = 0.121, *p* < 0.01, and β = 0.099, *p* < 0.001; respectively). We observed a full mediation effect of collective efficacy factors between organizational justice and subjective well-being.

## 4. Discussion

According to the literature, both organizational justice and collective efficacy are variables that predict good results at the social-organizational and individual levels [27,30,53,54,55]. This article aimed to explore the effect of organizational justice and collective efficacy on the subjective well-being of Chilean teachers. In addition, the association between these variables was analyzed considering sociodemographic variables such as age and gender and school context variables such as school SES.

The findings of this study are comparable to those of studies conducted in other countries when it comes to the link between the constructs [56,57,58,59,60,61]. There is broad agreement that a number of sociodemographic factors may have an impact on subjective well-being. Some studies reveal that between adolescence and adulthood, subjective well-being declines with age [62]; however, other studies show the contrary [63,64,65]. In the case of teachers, a positive relationship between age and subjective well-being was established [66,67,68], and in this regard, our findings follow this pattern. In terms of school context, this study also echoes national and international research [69,70,71], stating that SES is positively related to well-being, being especially true in the case of teachers, giving sense to the interactionist models where well-being is the result of the interaction between personal and environmental factors [72,73]

The proposed mediation model showed that organizational justice has a significant total effect on subjective well-being, which is totally mediated by the two dimensions of collective efficacy. It is important to note that organizational justice has a higher effect on the opportunities factor than the challenges factor (both statistically significant), suggesting that it might be part of the conditions that could promote collective efficacy in teachers [74,75]. To overcome daily challenges at school, it is important to have the support of the school management but also of peers [76].

The question that arises is how to promote collective efficacy in teachers when the educational context is increasingly challenging, not only because of the generation gap with students that continues to widen but also because of how the school context has been changing within the context of the COVID-19 pandemic. The answer to this is not simple, as concern for the mental health and subjective well-being of members of the educational community is critical in today’s world.

## 5. Conclusions

The results showed that organizational justice has a significant overall effect on subjective well-being. This relationship is fully mediated by the two dimensions of collective efficacy. This allows us to conclude that perceptions of collective efficacy are central to explaining well-being as intrinsic factors.

The present results lead us to rethink the type of predictors that we are considering for subjective well-being, where they are no longer only internal to the subject or their immediate and mediate environment but also consider organizational aspects that may be influencing. Among the projections of this study is to consider more contextual variables in this relationship, such as characteristics of the institution beyond the socioeconomic level and vulnerability and characteristics of the director so as to monitor these variables over time. Likewise, analyzing this phenomenon in different contexts would allow us to know if these results are generalized or if differences can be observed due to idiosyncratic characteristics.

The results show that factors such as belonging, good treatment, and human relations within educational institutions may be key to explaining the subjective well-being of Chilean teachers, and therefore central and local governments should make more efforts to promote the mental health of teachers since their contribution to the education and development of the country is of vital importance in a world that is increasingly competitive and prey to external factors such as epidemics, climate change, and armed conflicts.

Along with aiming at wage and time improvements for better peer interaction and organization, the new laws to improve the teaching career in Chile should also strengthen the culture of success, accountability, and commitment of the teacher as a worker [77,78]. This would create a positive feedback loop where collective effectiveness would improve outcomes and personal satisfaction while also benefiting the system’s users, such as students and other members of the educational community.

International experiences have demonstrated the robust relationship between organizational justice, job satisfaction, and the positive behavior of individuals [79]. In this sense, beyond the organizational cultures inherited from colonial times or implemented in more recent management models, educational institutions should strive to have more pleasant work environments to keep their teaching teams cohesive and remain competitive in a labor market where the skills of teachers are appreciated in other industries.

Collective efficacy and school climate have a proven association with respect to mental health and, thus, the subjective well-being of teachers [40]. Good treatment of these constructs can even lower absenteeism due to depression which is one of the psychological strains that may signify a maladaptive response to stressors, one of them being the sense of organizational justice [80,81].

Finally, these results could be of use in countries with similar accountability systems or those implementing innovations in terms of human resources. They can learn from the Chilean experience in the sense that a system based on organizational justice may have a positive impact on teachers’ quality of life, given that a happier person is also a more productive one [82]. There are pending challenges, such as coping with a shortage of specialists in some educational areas and more incentives so that new generations may be more interested in enrolling in teaching programs. In this sense, policymakers should keep in mind that improvements in developing economies must go hand in hand with improvements in education quality [83]. A career with better salaries and better well-being could be more inviting to join.

## 6. Limitations

This project had some limitations. First off, because the research was cross-sectional in nature rather than longitudinal, the conclusions drawn from it may not be entirely generalizable to different contexts. Second, the schools’ involvement was optional, which may have led to bias, and self-report forms were employed to collect data from the participants. In addition, it is important to consider as a limitation that all of them are self-report measures, and there are no measures different from this one, so they should be considered for a future study in order to take over the common method variance.

## Figures and Tables

**Figure 1 ijerph-19-10963-f001:**
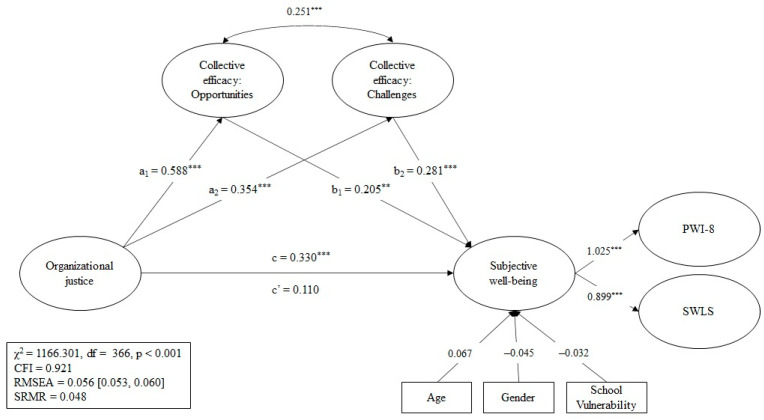
Mediation model using the pooled sample. *** *p* < 0.001. ** *p* < 0.01.

**Table 1 ijerph-19-10963-t001:** Descriptive Statistics, correlation matrix, and average variance extracted (AVE) for study variables (*n* = 693).

Variables	Measures	1	2	3	4	5
SWB	1. PWI-A	0.478				
	2. SWLS	0.796 ***	0.685			
Collective efficacy	3. Opportunities	0.316 ***	0.286 ***	0.456		
	4. Challenges	0.295 ***	0.272 ***	0.242 ***	0.279	
	5. Organizational justice	0.309 ***	0.278 ***	0.473 ***	0.267 ***	0.569
	M	8.39	8.31	4.91	5.07	4.86
	SD	1.25	1.54	0.81	0.72	1.04
	McDonald’s omega	0.887	0.914	0.769	0.609	0.878

Note. *** *p* < 0.001. Average variance extracted (AVE) appears on the diagonal.

## Data Availability

The datasets generated and/or analyzed during the current study are available from the corresponding author upon reasonable request.

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
