# Peer review of "Perceived Collective School Efficacy Mediates the Organizational Justice Effect in Teachers’ Subjective Well-Being"

_ijerph, 2022, doi:10.3390/ijerph191710963_

Round 1

Reviewer 1 Report

Thank you very much for submitting to the International Journal of Environment Research and Public Health. This is an interesting and important study that assesses the mediating role of perceived collective school efficacy in the relationship between organizational justice and teachers’ subjective wellbeing in a sample of 693 teachers in 16 urban regions of Chile.

Strengths:

This study has several merits. The authors were familiar with the literature, and the arguments were logical and easy to follow. The findings and theoretical implications were clearly presented and discussed.

Concerns:

The statistical analysis can be further improved if the following issues could be addressed further:

1.     The study was cross-sectional; therefore, alternative SEMs might exist (e.g., teachers with a higher level of SWB might also have a higher level of perceived organizational justice and collective efficacy). Have the authors considered examining those alternatives?

2.     Following 1, due to the cross-sectional nature, how would the authors plan to examine and control for the Common Method Variance (e.g., Podsakoff, P. M., MacKenzie, S. B., Lee, J. Y., & Podsakoff, N. P. 2003. Common method biases in behavioral research: A critical review of the literature and recommended remedies. Journal of Applied Psychology, 88 (5): 879–903.)?

3.     What were the changes/improvements in model fitness between the ‘null’ model (i.e., the one without a mediator) and the hypothetical model?

4.     Since bootstrap was used, is it correct to assume that there was no missing data? That is, EVERY respondent had answered EVERY survey question? If not (as indicated by the degrees of freedom of the SEM), what was the imputation method? And why?

5.     The sample contained school teachers. In addition to the information provided, which type(s) of schools did the teachers come from? E.g. primary school? High school? What were the proportions of these different types of schools? Will they have any implication for the findings?

6.     P.2 said that the SWB measure had seven dimensions of satisfaction, two additional items on religion/spirituality and one overall life satisfaction. Why then was it a 9- instead of 10-item scale?

7.     Before running the mediation SEM, it is suggested that a CFA is performed to examine whether the hypothesized latent variables are distinctive.

I hope the above comments may offer some help in improving the manuscript.

Reviewer 2 Report

Justification of the control variables used could be provided.

The references support the methods used, although it would be important to have provided more information about the concrete scales used, namely the scale to assess the "subjectivity weel-being".

There are no references to the limitations of the study, particularly with regard to the sample used and the context of other regions and professional activities.

The conclusions support the results presented, although it must have been mentioned that the same study should be carried out in other contexts so that the conclusions can be generalized.

I have not identified structuring aspects that require a mandatory review of this paper.

However, it is important that it is clarified what is the PWI-8 scale in Table 1, since this designation is not used in Section 2.2.1 concerning the scale used to assess the SWB.

It is suggested to introduce an explanation of the acronyms SWB in line 127 and the acronym OJ in line 154.

I wonder if in line 89 "Personal Well-being Inventory"  should be replaced by "Personal Well-being Index".

Reviewer 3 Report

Thank you for letting me review your work.

This is an interesting area of research, but overall I felt the results and discussion were very limited so did not do the work justice. More context of the work for a national and international audience would be useful, with more needed about what you are adding to previous work.

Please see further comments below:

·         Please review sentences for word duplication and amend, if necessary. An example is line 34-36 where ‘to this end’ is repeated.

·         The introduction may benefit from a brief introduction into education in Chilean schools to support an international readership. Is there anything that needs to be noted about education in Chile that led to this study being completed?

Methods:

·         Who was your target audience? Were the teachers targeted from primary or secondary schools, or both, or do schools in Chile target all year groups? See comment above about Chilean context for the reader

·         Lines 80-84 are results I believe so should be in the results section. The methods should contain information about your target sample size, not what was received.

·         More information is needed about distribution and collection of the survey. How was it sent to schools? How was it collected? Were reminders sent? How was data stored prior to analysis?

·         When was the survey completed?

·         More explanation about why the specific research instruments were chosen would be useful.

·         Line 93 – 9-items should read 9-item

Results:

·         This section would benefit from more analysis by variable. Currently there are limited findings presented.

·         More breakdown into analysis by variable e.g. gender, age, would be useful for the reader.

·         Was analysis by school size investigated?

Discussion:

·         This section was very limited. More detail is needed throughout to explain the results seen.

e.g. lines 166-168 are very generic.

·          I would have liked more detail as to why you believe there were differences by age and gender etc. do your results echo previous results, or are they different.

·         Are there any reasons as to why you think these results were seen? Were they what you expected?

·         Lines 175-181 look ahead. Is there any future work that needs to be done, or are there any solutions that could be suggested?

·         There is no mention of limitations in your research. Limitations needs to be noted.

·         How are these findings relevant for an international audience?

Reviewer 4 Report

Thank you for reaching out for the purpose of processing the assessment for the submitted contribution entitled "Perceived Collective School Efficacy Mediates Organizational 2 Justice Effect in Teachers' Subjective Wellbeing". I will list my observations in bullet points for greater clarity.

• A number of authors' literary sources are not current (older than ten years). Consider whether these resources are really necessary for the post.

• I'm missing the research limits chapter. The authors certainly see certain limits in their work and it is therefore appropriate to mention them.

• I have no serious reservations about the introductory part.

• The research tool is composed of three parts, each of which has several sub-dimensions. I am missing a reliability calculation for each of these parts.

• The actual model in Figure 1 should be described in more detail, not in terms of quantitative indicators (everything is fine here), but in terms of meaning for the reader.

• The discussion part needs to be elaborated more and the benefit of the study should be mentioned in more detail. The same goes for the conclusion.

• Complete the research questions and objectives.

Round 2

Reviewer 1 Report

Thank you very much for resubmitting the manuscript. There is clear evidence that the authors attempted to respond to the previous comments and suggestions. The manuscript shows improvements. There are a few issues that I wish the authors to address.

Major issues:

1. The response letter says that "our study relied on self-report measures and therefore does not have appropriate measures to test for common method variance". This unfortunately is not fully justifiable. Following the single unmeasured latent factor approach by Podskoff et al. (2003), the common method variance/bias can be examined.

2. It says that data imputation was performed (via regression). Before conducting data imputation, was the pattern of missing data examined? What was the percentage of missingness in each variable? Was that missingness (completely) at random or not?

3. Each multi-item scale's average variance extracted (AVE) should be reported (please see Fornell and Larcker, 1981).

4. What was the statistical power of the SEMs?

Minor issues:

1. P3 says that "87.0% of the teachers belonged to primary education…while 87% belonged to secondary education (15-18 years)". These percentages (85%) were confusing.

2. P3 says that "second-order factor for the Subjective well-being variable was tested considering PWI-A and SWLS)." – Please remove the “)”.

3, Variable labels should be consistent. For example, PWI-A was shown as PWI-8 in the graph.

I hope these could help the authors to improve the manuscript further.

References

Podsakoff, P. M., MacKenzie, S. B., Lee, J. Y., & Podsakoff, N. P. 2003. Common method biases in behavioral research: A critical review of the literature and recommended remedies. Journal of Applied Psychology, 88 (5): 879–903

Fornell, C., & Larcker, D. F. (1981). Evaluating structural equation models with unobservable variables and measurement error. Journal of Marketing Research, 18(1), 39–50.

Reviewer 3 Report

Thank you for addressing all my previous queries. The paper reads much better now. Good luck for future work.

Author Response

Thank you for your time. Your comments helped us to improve our work.